# Adverse Drug Reaction Reporting by Patients in 12 European Countries

**DOI:** 10.3390/ijerph18041507

**Published:** 2021-02-05

**Authors:** Agne Valinciute-Jankauskiene, Loreta Kubiliene

**Affiliations:** Department of Drug Technology and Social Pharmacy, Lithuanian University of Health Sciences, Sukileliu Ave. 13, LT-50166 Kaunas, Lithuania; loreta.kubiliene@lsmu.lt

**Keywords:** adverse drug reactions, direct reporting, promoting, pharmacovigilance

## Abstract

Patients who report suspected adverse drug reactions (ADRs) help minimize drug safety risks and bolster the pharmacovigilance system. The aim of this study was to examine the contribution of patients to pharmacovigilance and compare the tools used to promote patient reporting in European countries that implemented this reporting type in 2012–2013. A web-based questionnaire was sent to the national competent authorities (NCAs) of the European countries. The received answers were systematized and compared using statistical analysis. The performed statistical analysis demonstrated that changes in the number of received ADR reports increased significantly in each country during the analyzed period. These changes were significantly different in Ireland and Finland from those in the other reviewed countries. The common source of information on direct patient reporting was the country’s NCA website. Other sources used were social media pages, leaflets, and posters. This is the first study on patient reporting schemes implemented after the significant reform of the European regulatory system for pharmacovigilance. However, some countries did not actively promote their patient reporting schemes. Our findings indicate that countries with minimal experience in pharmacovigilance systems that include direct patient reporting should organize comprehensive campaigns on ADR reporting.

## 1. Introduction

The primary source of information about drug safety is health care professionals. However, studies have shown that doctors provide incomplete adverse drug reaction (ADR) reports or do not provide them at all [1]. The failure to provide adequate ADR reports by doctors is attributed to time constraints. Additionally, the direct patient reporting of ADR has been shown to provide information about ADRs from other perspectives. The combined information about ADRs from healthcare professionals and patients has a significant impact on signal detection of new, rare, or serious ADRs [2]. The amendments to the European Parliament Directive 2001/83/EC on the Community code relating to medicinal products for human use (Directive 2010/84/EU) obligate the Member States to take all necessary measures to encourage all parties within the health care system, including patients, to report suspected ADRs. The Member States should develop and provide reporting formats for direct patient reporting [3]. The significance of adopting such an improved reporting system is that patients can actively participate in critical health care decisions.

In several countries, the ADR reporting schemes for consumers have been promoted since the implementation of their pharmacovigilance systems or shortly after. For instance, the practice has been carried out in the US, New Zealand, the Netherlands, Denmark, and Sweden. After the introduction of new European pharmacovigilance legislation in 2012–2013, 15 countries in Europe implemented schemes for direct patient ADR reporting to competent authorities [4,5]. Some studies have investigated the motives of patients reporting ADRs, as well as patient reporting schemes within and outside of Europe. Additionally, studies have focused on the methods used to promote ADR reporting and sociodemographic and economic features as explanatory factors for population ADR reporting [2,6,7,8]. However, little research has been undertaken to examine the more recently developed patient reporting schemes implemented after 2012.

The aim of this study was to examine the contribution of patients to pharmacovigilance and compare tools used to promote patient reporting in European countries that implemented this type of reporting in 2012–2013. A web-based questionnaire and statistical analysis were used for achieving this aim.

## 2. Materials and Methods

A questionnaire was emailed to the national competent authority (NCA) of each eligible country. A country was eligible to participate in the study if they first implemented their direct patient reporting of ADRs in 2012–2013 in response to the new European pharmacovigilance legislation. In total, 15 countries were eligible and were sent the study questionnaire: Austria, Belgium, Bulgaria, Estonia, Finland, Germany, Greece, Ireland, Latvia, Lithuania, Luxembourg, Poland, Portugal, Slovakia, and Spain. The questionnaire was designed to collect statistical data (i.e., the number of patient-specific ADR reports in the first year of implementation and in 2017–2018), methods for patients to report ADR, tools in use to promote patient reporting, and whether personalized feedback about the reported ADR was routinely given to the patient.

A statistical analysis was performed to investigate the quantitative findings from different countries and provide a numerical estimate of significance of the change during the analyzed period in the interviewed countries. The analysis was performed on the number of patient-specific ADR reports submitted to competent authorities for the first year direct patient reporting was implemented and the most recent year (2017 or 2018). The comparison was completed using a random-effects model and fixed-effects model to evaluate the changes in the number of received ADR reports in participating countries during the analyzed period. Confidence intervals (95%) for each data set were calculated. MedCalc software (version 12; MedCalc Software Ltd., Ostend, Belgium) was used for data analysis. Heterogeneity was explored using the Cochran Q-test of heterogeneity. The significance was determined by reviewing the location of odds ratios. An odds ratio of one indicates no effect/no difference between interventions. Any line which crosses the line of null effect does not illustrate a statistically significant result. Countries with narrow confidence intervals (horizontal lines) and crossing one (vertical line) are inconclusive. Countries with narrow confidence intervals that do not cross one are judged statistically significant. If the odds ratios with confidence intervals fail to overlap, the differences between the two countries are statistically significant. If the results are similar between various countries, the data is homogeneous and statistically insignificant.

## 3. Results

Of the 15 countries that were contacted, 12 responded to the questionnaire. Follow-up emails were sent out to the nonresponsive NCAs—Luxembourg, Poland, and Bulgaria—but their responses were not received. The answers from 12 countries were representative of 80% of all eligible countries, which is a substantial portion for conducting a comprehensive study.

In the participating countries, the primary methods used for ADR patient reporting were direct mail, email, phone call, fax, and online internet-based methods (Table 1). These methods were available in seven (58%) countries: Estonia, Austria (phone call only in addition to written form), Ireland, Belgium, Lithuania, Germany, and Greece. However, the online form of reporting was not available in Finland; email and phone calls were not available in Latvia; and fax was not available in Finland, Portugal, and Slovakia. The ADR reporting system in Spain was decentralized, so reporting methods could differ by region. However, an online reporting method was available at the national level. Of the 12 participating countries, 10 (83%) provide specific patient forms that differed from the forms intended to be used by healthcare professionals. Only two countries (16.6%) (Finland and Ireland) in our study currently did not offer patient-specific ADR reporting forms.

Personalized feedback was given to patients who reported ADRs in five (41.6%) countries: Estonia, Austria, Portugal, Spain, and Slovakia. In the other countries, feedback was provided with some exceptions. For example, in Belgium, feedback was given only if the patient’s report included questions. In Portugal, all reporters received a confirmation of submission. Additional information was requested from patients when necessary. Furthermore, causality was assessed in the case of serious ADRs in Spain and reported to patients. For the rest of the countries, patients who used the online reporting form received an automatic letter of confirmation upon submission.

All participating NCAs had webpages where information about ADR reporting and standardized information could easily be found. Other methods used to promote patient reporting schemes are listed by country in Table 2. Besides webpages, the most common methods were social media pages (e.g., Facebook), leaflets, and posters. Ireland and Portugal actively collaborated with patient associations and provided information about ADR reporting to targeted populations. However, three (25%) countries—Latvia, Spain, and Finland—did not actively promote their patient reporting schemes, which were limited to the information provided on the NCA webpages. Germany did not name their promotion tools, though the information could be found on their NCA webpage.

Statistical analysis was performed for the comparison of received ADR reports during the period from the beginning of patient reporting schemes until the 2017–2018 period. Because the total number of received ADR reports was not provided by its NCA, Greece was excluded from the statistical analysis. The results of the statistical analysis and related forest plot are presented in Figure 1.

The calculated results indicate that the number of direct patient reports changed significantly in each individual country during the analyzed period. Additionally, significant differences between some countries were observed. Furthermore, our findings indicate a relationship between a high number of different promoting tools used and increased patient ADR reporting. As shown in Figure 1, the plots for Finland and Ireland lie on opposite sides of the statistical analysis plot. The rise in ADR reports from patients was significant in both of these countries. However, the observed increase in Ireland was higher. The tools used in Ireland to promote ADR reporting included close engagement with patient groups and targeted awareness campaigns. By contrast, Finland promoted direct ADR reporting by informing patients on the NCA webpage only. Therefore, the increase in reports could be related to factors other than promoting tools. This assumption could also be applied to Latvia and Spain, both of which only promoted patient ADR reporting on the NCA webpage.

## 4. Discussion

All the countries that participated in this study started patient ADR reporting in 2012 or 2013 after the major reform of the European regulatory system for pharmacovigilance was implemented in July 2012 [3]. The number of different publications brought out some fundamental conclusions. First, patient reporting of ADRs provides a valuable contribution to pharmacovigilance by increasing the number of reported ADRs, thus favoring the possible detection of ADRs at the early stages of reactions. After the implementation of a consumer reporting system, the first step is to raise awareness among the general public about the possibility of adverse reactions and the importance of reporting them, and to actively encourage consumers to contribute to drug safety [9,10,11,12,13,14]. The present study examined the number of patient-specific ADR reports received during the first year of implementing consumer reporting schemes in 12 European countries. Additionally, the tools used to promote patient ADR reporting were explored.

Based on the number of received ADR reports for the 12 participating countries, the problem of underreporting ADRs was seemingly present. However, it is not easy to establish an accurate measure of the level of underreporting in these countries. However, based on the findings of this study and existing research, the levels of underreporting are estimated to reach 90% or more [15]. Although the reasons for underreporting by health care professionals, pharmacists, and consumers were not analyzed in the present study, several studies have documented potential causes, including lack of time, other priorities, and lack of awareness [16,17,18,19]. In contrast to those faced by health care professionals, the main barriers to reporting ADRs for consumers are poor awareness, lack of knowledge about who should report and to whom, difficulties with reporting procedures, the high costs involved, and a lack of feedback [6].

The success of direct patient reporting depends on the adequate knowledge of pharmacovigilance systems and the tools used to inform society about using them [4]. The methods used to educate patients about reporting ADRs varied among the participating countries. For some countries, only a few tools were applied, including leaflets, posters, or the NCA webpage only. The countries that participated in this study had five or six years of experience in direct patient reporting. As a result, it is essential for countries with minimal experience to organize comprehensive campaigns on ADR reporting. Discussions about organizing such campaigns should include key stakeholders including general practitioners, pharmacies, patient organizations, and competent authorities. The lack of active forums and channels for promoting the direct patient reporting scheme supports the results of previous studies, which have indicated that a negative attitude towards new pharmacovigilance systems dominates among NCAs. Patient reporting is considered to be a regulatory mandate, but it is not regarded as a supporting tool for drug safety [2,5,6]. The results of this study showed that the methods used to inform patients about reporting ADRs have an essential impact. Additionally, a significant increase in ADR reports were found in each country during the studied period, with Ireland showing the greatest growth.

The primary goal of the EU pharmacovigilance system is to deliver safer and faster decisions in medicine and health care. A systematic review showed that the median interval between the first reported ADR and the withdrawal of a drug launched after 1960 is three years, which is two times shorter than for drugs launched before 1960 [20]. This shortened period indicates that ADR signal detection and the regulation of new medicines improved; however, ADRs remain a significant health issue worldwide. Spontaneous ADR reporting by patients has been shown to be a valuable tool for early safety signal detection [13,21]. NCAs need to focus on educating patients to become active participants in pharmacovigilance management. Additionally, considering the significant roles that new technologies play in everyday life, NCAs should include the use of e-technologies to promote patient ADR reporting rather than only providing static information about ADRs.

Important data has shown in several studies that the active pharmacovigilance approach implemented in healthcare can significantly increase the number of ADRs [22,23]. Active calls to patients or reminder messages help to draw consumers’ attention to possible ADRs that might be mistakenly assigned to illness instead of the medicine in use [23,24]. Active pharmacovigilance might help develop consumer habits of reporting ADRs in the presence of spontaneous pharmacovigilance; therefore, the active approach needs to be considered, at least for some groups of medicines.

Our research has some limitations. First, not all invited competent authorities participated in the study or provided requested information. Secondly, the data does not show the changes that have taken place each year. The promoting tools were implemented by each NCA gradually. Statistical data was not collected for each year; therefore, the impact of separate tools on ADR reports cannot be measured. Despite these limitations, the information is still useful for policymakers and national authorities responsible for ADR report collection.

## 5. Conclusions

This is the first study that summaries ADR reporting by patients in European countries that recently implemented patient reporting schemes. The number of ADR reports received from patients by competent authorities and the variety of methods used to promote direct patient reporting indicates a lack of adequate knowledge on the pharmacovigilance system and an insufficient level of health education among the public. Going forward, it is critical for health care practitioners, in collaboration with health care agencies, to establish concrete and reliable systems that can be used to help patients report ADRs. One formidable means of achieving this objective is to maximize public awareness of the importance of reporting ADRs to health care providers and its positive potential for improving the overall well-being of society.

## Figures and Tables

**Figure 1 ijerph-18-01507-f001:**
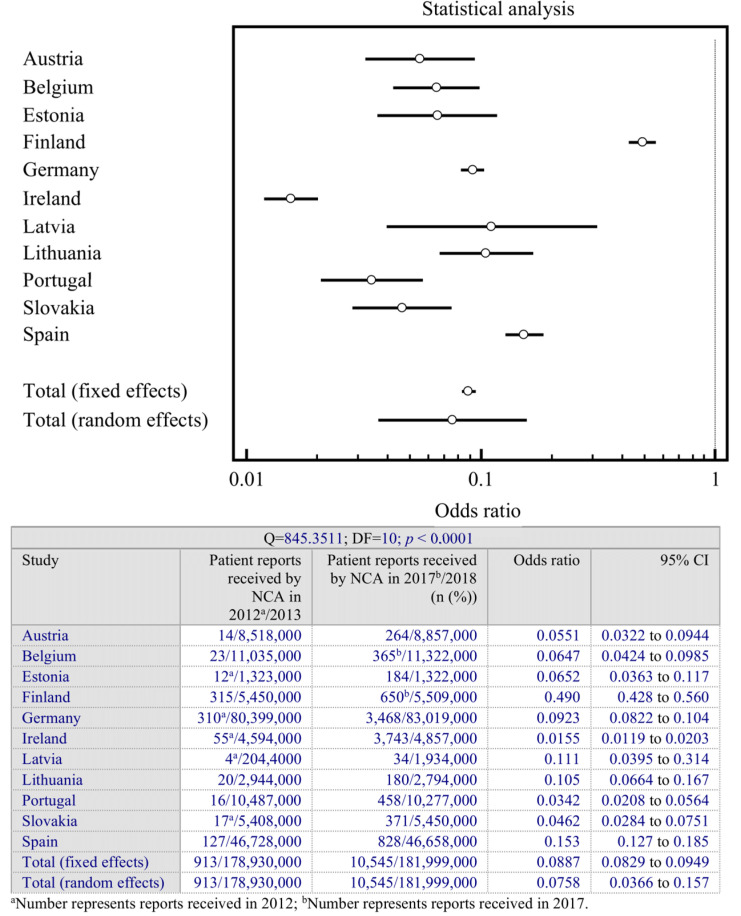
The changes in the number of ADR reports received individually by each country and overall.

**Table 1 ijerph-18-01507-t001:** Overview of ADR patient reporting in participating countries.

Country	NCA Website	Specific Patient Reporting Form	Methods of ADR Report Collection	Personalized Feedback
**Austria**	http://www.basg.gv.at	Yes	Direct mail, email, phone call (only as an addition to written form), fax, online	Yes
**Belgium**	http://www.fagg-afmps.be	Yes	Direct mail, email, phone call, fax, online	No, unless a question is asked
**Estonia**	http://www.sam.ee	Yes	Direct mail, email, phone call, fax, online	Yes
**Finland**	http://www.fimea.fi	No	Direct mail, email, phone call	No
**Germany**	http://www.pei.de http://www.bfarm.de	Yes	Direct mail, email, phone call, fax, online	Only receive an acknowledgement, no assessment
**Greece**	http://www.eof.gr	Yes, in the yellow card online. No, in the printed form.	Direct mail, email, phone call, fax, online	As applicable
**Ireland**	http://www.imb.ie	No	Direct mail, email, phone call, fax, online	No
**Latvia**	http://www.zva.gov.lv	Yes	Direct mail, fax, online	Not personalized feedback/standard text
**Lithuania**	http://www.vvkt.lt	Yes	Direct mail, email, phone call, fax, online	No, unless additional information is required
**Portugal**	http://www.infarmed.pt	Yes	Direct mail, email, phone call, online	Yes, all reports receive a confirmation of submission and additional information requests when necessary, and causality assessment in case of serious ADRs
**Slovakia**	http://www.sukl.sk	Yes	Direct mail, email, phone call, online	Yes
**Spain**	http://www.aemps.gob.es	Yes	National agency has an online form, but methods may vary by region due to a decentralized system	Yes, if the online form is used

Abbreviations: ADR, adverse drug reaction.

**Table 2 ijerph-18-01507-t002:** Adverse drug reaction (ADR) reports submitted by patients in the participating countries.

Country	Promoting Tools	Patient Reports Received by NCA in 2012 ^a^/2013 (n (%))	Patient Reports Received by NCA in 2017 ^b^/2018 (n (%))
**Austria**	Advertising, leaflets	14 (0.28%)	264 (2.6%)
**Belgium**	Annual cross-European SCOPE campaign in November	23 (0.4%) ^a^	365 (6.6%) ^b^
**Estonia**	Social media (Facebook), articles in newspapers, NCA webpage	12 (5%) ^a^	184 (36%)
**Finland**	NCA webpage	315 (14%)	650 (21%) ^b^
**Germany**	N/A	310 (0.6%) ^a^	3468 (16%)
**Greece**	PhD campaigns (NCA webpage, media, social media, interaction with patient organizations, etc.)	4.9%^a^	23. (9%)
**Ireland**	Public awareness campaigns, engagement with patient groups on specific safety issues and via the IPPOSI platform, provision of targeted publications in relation to medicines safety and ADR reporting	~55; 2% ^a^	~3743 (36%)
**Latvia**	NCA webpage	4; 1% ^a^	34 (5%)
**Lithuania**	NCA webpage, social media (Facebook), publications	20 (5%)	180 (12%)
**Portugal**	Communication campaigns, leaflets, posters, infographics, all of which are disseminated in social media in order to achieve patient recruitment, training sessions for promoting reporting in collaboration with patient associations, and articles in newspapers and magazines dedicated to the general public	16 (1%)	458 (5%)
**Slovakia**	A collaboration with webpage Slovak patient (https://www.slovenskypacient.sk/), planned publications	17 (1.8%) ^a^	371 (38.8%)
**Spain**	NCA webpage	127 (1%)	828 (6.8%)

^a^ Number represents reports received in 2012; ^b^ Number represents received reports in 2017. Abbreviations: IPPOSI, Irish Platform for Patient Organisations, Science, and Industry; NCA, national competent authority; SCOPE, the Strengthening Collaboration for Operating Pharmacovigilance in Europe.

## Data Availability

The data presented in this study are available on request from the corresponding author. The data are not publicly available due to the confidentiality policy.

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
