# Peer review of "Adverse Drug Reaction Reporting by Patients in 12 European Countries"

_ijerph, 2021, doi:10.3390/ijerph18041507_

Round 1
Reviewer 1 Report
The article is an interesting paper on direct patient reporting of ADRs in 2012-2013 in response to the new European pharmacovigilance legislation and after 5 year from the legislation. The article is interesting, I enjoyed reading it, and showed a substantial increase in ADRs signaling in time.
The limitations of the study regard mainly the fact that only 15 countries of the EU where eligible, and only 80% of them responded to the queries.
Another limitation is surely the fact that, although all countries than a patient reported ADRs increase, reaching almost 40% of all reported ARDs in some countries, these results do not seem always related to promoting tools, so it is very difficult to do any confront among countries.
I have some queries:
- When you report a program for statistical analysis, please report also the manufacturer and its city; for example: (MedCalc Software Ltd,Ostend,Belgium)
- In the discussion part,(page 6 line 190) when you talk about the fact that the median interval between the first reported ADR and the withdrawal of a drug launched after 1960 is three years, you should talk about the importance of active pharmacovigilance on new drugs, that is directly responsible of these results. I suggest you the reading of this article that highlights the importance of active pharmacovigilance for new drugs and that could be incorporated:" Iannone LF, Bennardo L, Palleria C et al. Safety profile of biologic drugs for psoriasis in clinical practice: An Italian prospective pharmacovigilance study. PLoS One. 2020 Nov 3;15(11):e0241575."
Thank You
Reviewer 2 Report
The manuscript discusses the evolution of ADR reporting by patients in 12 European Countries. The manuscript is well-written and the discussion is supported by data. However, the authors should deal with some major issues.
It is not clear how the 15 eligible countries were selected. if the selection was based on literature data (i.e. REF. 4), some discrepancies are on the field. Based on Herxheimer's manuscript, Finland, France, Germany, Ireland, Portugal, Spain did not actively collect patients reports, whereas other countries did. Despite this, Finland, Germany, Ireland, Portugal, and Spain were included, whereas others with a reporting system not (e.g. Italy). Moreover, the information reported in REF. 4 referred to 2010, not to 2012-2013. How did authors get the information about the existence of patient reporting systems in 2012-13? The authors should substantiate better the initial selection of countries.
The limitations of the study should be discussed.
Round 2
Reviewer 1 Report
The author responded to all queries. Article is ready to be published
Reviewer 2 Report
The manuscript may be considered acceptable for publication.